# Cardiac-Gated Neuromodulation Increased Baroreflex Sensitivity and Reduced Pain Sensitivity in Female Fibromyalgia Patients

**DOI:** 10.3390/jcm11206220

**Published:** 2022-10-21

**Authors:** Kati Thieme, Kathrin Jung, Marc G. Mathys, Richard H. Gracely, Dennis C. Turk

**Affiliations:** 1Department of Medical Psychology, Philipps-University Marburg, 35037 Marburg, Germany; 2Center for Pain Research and Innovation, University of North Carolina, Chapel Hill, NC 27599, USA; 3Department of Anesthesiology and Pain Medicine, University of Washington, Seattle, WA 98195, USA

**Keywords:** neuromodulation, baroreflex sensitivity, dmNTS, pain inhibition, fibromyalgia

## Abstract

The study presents a novel approach of programing pain inhibition in chronic pain patients based on the hypothesis that pain perception is modulated by dysfunctional dorsal medial nucleus tractus solitarii (dmNTS) reflex arcs that produce diminished baroreflex sensitivity (BRS) resulting from a conditioned response. This study tested whether administration of noxious and non-noxious electrical stimuli synchronized with the cardiac cycle resets BRS, reestablishing pain inhibition. A total of 30 pain-free normotensives controls (NC) and 32 normotensives fibromyalgia (FM) patients received two, ≈8 min-epochs of cardiac-gated, peripheral electrical stimuli. Non-painful and painful electrical stimuli were synchronized to the cardiac cycle as the neuromodulation experimental protocol (EP) with two control conditions (CC1, CC2). BRS, heart-rate-variability (HRV), pain threshold and tolerance, and clinical pain intensity were assessed. Reduced BRS in FM at baseline increased by 41% during two, ≈8 min-epochs of stimulation. Thresholds in FM increased significantly during the experimental protocol (all Ps < 0.001) as did HRV. FM levels of clinical pain significantly decreased by 35.52% during the experimental protocol but not during control stimulations (*p* < 0.001). Baroreceptor training may reduce FM pain by BRS-mediated effects on intrinsic pain regulatory systems and autonomic responses.

## 1. Introduction

There is little consensus regarding the mechanisms underlying persistent pain in individuals diagnosed with fibromyalgia (FM) [1]. Several studies have identified distinct subgroups based on biopsychosocial factors. This heterogeneity suggests that different functional mechanisms may be involved [2,3,4]. Two studies by our group [3,5] reported that two major subgroups of FM are characterized by altered blood pressure (BP). Of the samples, 46.7% revealed an elevated BP pattern in response to stressful stimuli delivered in a laboratory setting [3].

BP elevations influence baroreceptors, that, in turn, provide input to the dorsal medial nucleus of the solitary tract (dmNTS) reflex arcs that modulate BP, heart rate (HR), vessel dilation, autonomic balance, sleep, and pain perception (Figure 1) [6,7,8,9,10]. The dmNTS projects via excitatory (glutamatergic) fibers in the ascending reticular activating system (ARAS) to the rostral ventrolateral medulla (RLVM), coordinating multi-phasic autonomic responses throughout the body (e.g., [11]). The dmNTS modulates brain network regions such as the amygdala and anterior cingulate cortex that are critically involved in pain perception and sleep (e.g., [12,13,14,15,16]).

In normotensive individuals, increasing BP stimulates carotid sinus aortic baroreceptors, which, in turn, decreases sympathetic tone and increases analgesic parasympathetic activity [17] by the inhibition of activity in the ARAS, a non-specific cortical projecting system (e.g., [18,19,20]). In pain-free normotensive individuals there is an inverse relationship between BP and pain intensity [6,18]; thus in healthy individuals, as BP increases pain decreases. In contrast, in FM, orofacial pain, migraine, rheumatoid arthritis, and baroreflex sensitivity (BRS) is reduced and the responses to stress and pain are blunted [7,18,21]. The inverse relationship between BP and pain is abrogated where higher BP yields increased pain perception [5,22]. Persistent pain in FM appears to be associated with impaired interaction between BP and pain sensitivity [5].

What causes this observed altered BP/pain relationship? One explanation is that chronic stress is associated with vagal withdrawal [23] producing a sympathetically mediated increase in arterial BP in individuals with hypertensive stress reactivity. Due to the inverse relationship of BP and stress, individuals experience a reduction in perceived stress. BP is negatively reinforced and increases over time as a learned hypertension [24] while the BP variability is decreased [8], resulting in diminished BRS and dmNTS activity. These effects tend to maintain pain chronicity [18,25]. The underlying mechanisms likely involve operant conditioning of BP under stress exposure in normotensive individuals with a predisposition for hypertension [26].

Early animal studies show reduced pain after electrical stimulation of NTS (e.g., [27], and in human studies after mechanical stimulation of baroreceptors in carotid sinus dependent on cardiac cycle [17], associated with the activation of vagal tone and pain inhibitory systems [28,29]. Thus, it appears that BRS is mediated by the NTS reflex arcs [30,31] and higher BP variability [32].

The current study examines the interaction between BP, BRS, and heart rate variability (HRV) as indicator of a sympathetic stress response as well as pain perception. 

We hypothesize that electrical stimuli delivered during systolic and diastolic phases of the cardiac cycle in a randomized order would reset BRS, regulate HRV and reduce pain sensitivity and clinical pain in a subgroup of FM patients characterized by normotensive resting BP.

## 2. Material and Method

### 2.1. Participants

Thirty-two female patients, who met the revised American College of Rheumatology criteria for FM [1], recruited at the University of North Carolina, were included in this study. Exclusion criteria consisted of hypotension, pain disorders caused by inflammation, neurological complication, concomitant severe disease, pregnancy, selective beta blockers intake, use of muscle relaxants or opioids, major psychiatric disorders, and lack of English language fluency. Thirty age- and sex-matched normotensive individuals served as control participants (NCs). FM patients and the female NCs were comparable in age and other demographics variables. An institutional review board approved the study, which adhered to the Declaration of Helsinki. Informed consent was obtained from all study participants.

### 2.2. Procedure

#### 2.2.1. Clinical Assessment

A rheumatologist diagnosed FM for all the patients included. The rheumatologist performed an examination and reviewed laboratory measurements (i.e., rheumatoid factor, antinuclear antibodies, erythrocyte sedimentation rate) on all FM patients. 

#### 2.2.2. Stimulation Protocol Rationale

A set of physiological observations and potential mechanisms contributed to the design of the experimental protocol (EP) that tested the primary hypothesis that gating sensory stimuli delivered during specific cardiac cycle phases will reset BRS and reduce clinical pain in subpopulations of FM patients characterized by normotensive resting BP: (1)Persistent increases in BP or hypertension are associated with reduced heart rate variability (HRV), BP variability, BRS, and chronic pain consistent with Dworkin’s theory of learned hypertension (15).(2)Arterial baroreceptors modulate processing of nociceptive input dynamically during BP changes within the cardiac cycle. Electrocortical activity and sensory perception vary with cardiovascular events that alter baroreceptor activity in individuals with the predisposition of hypertension (18, 19, 49).(3)Cardioinhibitory mechanisms regulate rapid changes in BP primarily by decreasing sympathetic activity and HR. This integrated process provides cardiac protection in response to rapid increases in BP by decreasing sympathetic outflow from the vasomotor center of the brainstem, and by increasing parasympathetic outflow from the nucleus ambiguus, and dorsal motor nucleus of the vagus nerve.

Enhancing BRS may not have a pain-inhibitory effect for all pain modalities possible due to varying anatomical pathways that contribute to pain perception and autonomic integration. There is evidence that tactile pressure and electrical pain stimuli preferentially activate BRS in contrast to thermal pain [33]. 

In addition to the potential physiological mechanisms described, a learning interpretation suggests that: (1)Phasic elevations in BP may engage endogenous coping mechanisms. Elevations in BP are usually a healthy and effective response to stress and pain. However, in chronic pain patients, the hypertensive response may persist in cases in which stress remains elevated, operantly conditioning elevated BP. To decondition this BP-mediated effect, the experimental protocol adjusts electrical stimulation current to evoke both painful and non-painful sensations during diastolic and systolic BP within the cardiac cycle. Varying stimulus intensities produces variable changes in BP values, and BRS increases [17,34].(2)Given that cortical activity is decreased during the systolic phase of the cardiac cycle, electrical stimuli should be synchronized with the cardiac cycle [17].(3)To induce cardio-inhibitory effects and to increase BRS, the electrical stimuli should be delivered immediately after the systolic and diastolic BP peaks in randomized order to prevent habituation.(4)Electrical stimuli are ideal because they can be precisely controlled and delivered during specific cardiac phases and can activate the BRS and induce cardio-inhibitory effects.(5)Before and after the stimulation, individual pain threshold and tolerance are re-evaluated and adjusted as necessary to (a) maintain meaningful painful and non-painful intensities and (b) to provide feedback to the participant with respect to reduced pain perception that positively reinforces the desensitization of pain perception.

#### 2.2.3. Stimulation Protocols

**Experimental Protocol (EP)**—In this protocol, participants experienced brief 250 ms stimulus trains of either non-painful sensory or painful (75% and 50% of the tolerance value) electrical stimuli, ordered randomly, during either the systolic or diastolic phase of the cardiac cycle in two epochs each lasting ≈8 min. A total of 66 trains of electrical stimuli were delivered, in various stimulus intensities (sensory, 50% and 75% of tolerance value) and cardiac cycle phases (diastolic, systolic). This resulted in 33 trains of stimuli delivered during each of the systolic and diastolic phases of the cardiac cycle.

**Control Condition 1 (CC1) Protocol**—This control protocol was comprised of two ≈8 min epochs with only painful electrical 250ms stimulus trains at 50% and 75% of individual pain tolerance, delivered during either the diastolic or systolic phase of the cardiac cycle. Because this experimental condition delivered only painful stimuli, it was used to examine the effects of associative or classical conditioning of pain inhibition. A total of 44 trains of electrical stimuli were delivered, in various stimulus intensities (50% and 75% of pain tolerance value) and cardiac cycle phase (diastolic, systolic). This resulted in 22 trains of stimuli delivered during the systolic and diastolic phases of the cardiac cycle.

**Control Condition 2 (CC2) Protocol**—In this control protocol, non-painful sensory and painful stimulus 250 ms stimuli train at 50% and 75% of the tolerance threshold values were delivered independent of the cardiac cycle in two, ≈8 min-epochs. A total of 66 trains of electrical stimuli were delivered similar to the EP protocol. Application of non-painful and painful stimuli independent of the cardiac cycle provided a sham condition that determined if cardiac gating contributed to the observed outcomes.

#### 2.2.4. Psychophysiological Assessment

FM patients and NCs were instructed not to consume any analgesic, antidepressant, or antihypertensive medication for four days prior to their scheduled psychophysiological assessment. A 35 min psychophysiological session to validate the normotensive resting BP and the hypertensive stress reactivity followed the medical and psychological assessments. 

Each of the three stimulation protocols consisted of two, 8 min stimulation periods, with stimulus intensity calibration before, between and after (Figure 2). BP and HRV were recorded throughout the session. The EP, CC1 and CC2 protocols were delivered in a counterbalanced order. 

Each stimulus train was delivered to electrodes connected to the index and ring finger of the right hand and each train was 250 ms in duration. Stimulus intensities corresponding to subjective levels of sensory detection threshold, pain threshold and pain tolerance were determined by administering an ascending series of trains beginning at 0.2 mA and increasing in 0.2 mA steps up to a maximum of 4.6 mA. Participants rated the sensations evoked by each stimulus on an 11-point scale (0–10) with verbal anchors of ‘no pain’ to ‘most intense imaginable pain’. The largest rating of ‘0’ was used to define the sensory threshold. The lowest stimulus intensity rated as either ‘1’ or ‘2’ was used for the electrical pain threshold, and ‘10’ rated stimuli rated were used to define the electrical pain tolerance. A calibration consisted of the mean result of 2 ascending series and computation of stimulus intensities that were either 50% or 75% of pain tolerance values. These specific values were chosen because other studies of operant conditioning in chronic pain have shown that pain stimuli that are higher than 50% can trigger positive reinforcement operant conditioning of pain inhibition (20). Additionally, since electro-cortical activity and sensory perception vary with cardiovascular events that alter baroreceptor activity (18,49,50), the randomized delivery of pain-free, 50% with 75% stimuli after systolic or diastolic peaks is comparable to an interval training of baroreceptors (Figure 2). This protocol was designed to provoke intermittent reinforcement of operant conditioning.

#### 2.2.5. Psychophysiological Recordings

Participants were seated and positioned in a straight back chair and were instructed to move as little as possible. All instructions were presented on a video screen. The presentation of the instructions, data acquisition, and data storage were computer-controlled. The following physiological measurements were recorded continuously: BP and HR in beats-per-minute [35] were monitored using a Finapres BP monitor attached to the middle finger of the left hand. (Finapres Medical Systems, Enschede, the Netherlands). A LabLinc V modular instrument (Coulbourn Instruments, Whitehall, PA, USA) was used to record electrocardiogram (ECG). A computer program averaged the sample time synchronized to the R-wave of the electrocardiogram. BRS was calculated as the average of the instantaneous ratio of BP and HR [36] during increasing BP sequences. 

HRV was evaluated by both ECG and beat-to-beat changes in BP ([37]; see the recommendations provided by the task force of the European Society of Cardiology and the North American Society of Pacing and Electrophysiology. 

*Time Domain Measures*: HR was measured as inter-beat-intervals and a reciprocal value was calculated to derive HR values as beats-per-minute (bpm). SDNN (standard deviation of normal-to-normal [N–N] intervals) is the standard deviation of cardiac cycle inter-beat intervals, measured in milliseconds (ms), and it reflects all cyclic components of the variability in the recorded series of inter-beat-intervals. RMSSD (root mean square of the differences between successive N–N intervals) is measured in ms and estimates high-frequency variations in heart rate in short-term recordings that estimate of parasympathetic regulation of the heart.

*Frequency Domain Measures*: Total Power (TP) is used as a short-term estimate of the total power of power spectral density in the range of frequencies between 0 and 0.4 Hz. This measure reflects overall (cardio-sympathetic and cardio-parasympathetic) autonomic activity. Very-low-frequency (VLF) is the power spectrum frequency band ranging between 0.0033 and 0.04 Hz. This measure is not so well defined in terms of physiological mechanisms; however, activity in this band has been associated with regulation of the renin-angiotensin system and is used as an indicator of activity of slow temporal processes regulated by the sympathetic nervous system. Low-frequency (LF) is a band of the power spectrum that ranges from 0.04 to 0.15 Hz and reflects both sympathetic and parasympathetic activity. It is an indicator of sympathetic activity in long-term recordings. Parasympathetic influence is reflected in the LF band when respiration rate is less than 7 breaths per minute. Thus, when a participant is in a state of relaxation with a slow and even breathing, LF values indicate parasympathetic activity rather than increased sympathetic regulation. LF band activity has also been related to BRS, with higher values associated with greater BRS [38,39,40,41]. Greater BRS means greater reflex changes in HR for a given change in mean arterial pressure (MAP) and is associated with enhanced baroreflex cardio-parasympathetic responses. High-frequency (HF) is a band of the power spectrum that ranges between 0.15 and 0.4 Hz and reflects parasympathetic activity. HF is known as the respiratory band because it corresponds to the inter-beat-interval variations as influenced by respiratory sinus rhythm. Slower and even breathing causes an increase in cardio-parasympathetic activity and the amplitude of the HF peak in the power spectrum. Frequency domain measures are calculated in milliseconds squared (ms^2^).

**Systole Stimulus Synchronizer.** In order to enable delivery of 250 ms electrical pulse trains that were phase locked to either the diastolic or systolic phase of the cardiac cycle, the University of North Carolina fabricated a device that assessed the R-wave from a 3-lead ECG signal on a beat-by-beat basis. This device, a Synchronizer, was used to trigger a constant current electrical stimulator (A13-75 Bioelectric Stimulus Isolator from Coulbourn Instruments, Whitehall, PA, USA) that delivered electrical pulses during specified points in the cardiac cycle. Consecutive inter-beat-intervals were used to dynamically calculate an average inter-beat-interval from 3 consecutive intervals. This average value was then used to deliver the electrical stimulus during the next beat in the cardiac cycle. The 250 ms pulse trains delivered during the systolic phase were delivered at 20% of the value of the average inter-beat-interval, whereas diastolic pulse trains were delivered at 80% of the value of the dynamically determined average inter-beat-interval (Patent: US9604054B2). 

### 2.3. Data Analysis

Biometric data analysis was performed in four sequential steps. 

The first step examined baseline differences in sensory, pain threshold, and pain tolerance in the FM and NC groups after outlier elimination (defined >2 standard deviations from the mean). Repeated measures analysis of covariance (ANCOVAs) evaluated the effect, controlling for baseline differences in the 3 threshold phases, as a within factor and the 2 groups as between factors in the EP, CC1, CC2 protocols and were followed by post hoc *t* tests. This post hoc analysis was used to calculate both group differences in each protocol and group differences in percent change of sensory, pain threshold, and tolerance in each protocol. To assess the efficacy of each protocol for increasing thresholds, the 3 thresholds after the 2nd epoch were compared for each group with paired-sample *t*-tests. 

The second step assessed changes in clinical pain before and after each protocol using a visual analogue scale (VAS 0-100), where 0 means ‘no pain’ and 100 ‘the worst imaginable pain’. 

The 3rd step assessed BL differences in BRS using *t*-tests for within and between group comparisons. 

The 4th step evaluated HRV variables: (total power (TP), high frequency (HF), low frequency (LF), very low frequency (VLF), standard deviation of normal-to-normal R–R (heartbeat) intervals (SDNN), root mean square differences of successive R–R intervals (RMSDD) in FM and NC and tested changes in each protocol using non-parametric tests for between (U-test) and within the group comparisons (Friedman test). Furthermore, the changes between baseline and each epoch were compared between the 3 different protocols. 

## 3. Results

In the following paragraph, we present the changes in (1) Thresholds, (2) Clinical Pain, (3) BRS, and (4) Heart Rate Variability.
**1.** **Changes in Thresholds**

Baseline tolerance threshold (TT), pain threshold (PT), and sensory threshold (ST): divergent from comparable ST in FM patients and the NC group, PT (F_1,69_ = 20.36, *p* < 0.001) and TT (F_1,69_ = 28.75, *p* < 0.001) differed significantly. FM patients showed a 25.85% lower TT value and a12.89% lower PT value than the NC group. 

Threshold changes in the three different protocols: in the EP protocol, the ANCOVA revealed a significant threshold x epoch x group x interaction for ST, PT and TT (F_4,107.88_ = 3.056, *p* = 0.018) with significant differences between FM and NC groups (*p* = 0.008). Compared to baseline, the NC group showed increases in PT and TT to 9.4% and 11.6% (both *p* values < 0.01) after two, ≈8 min stimulations whereas in the FM group PT and TT increased to 15.1% and 25.1%, respectively (both *p* values < 0.001, Figure 3). After two, ≈8 min stimulations using the EP protocol, pain threshold and tolerance values in FM were significantly higher than those assessed after the CC1 and CC2 protocols (all *p* values < 0.05).

For the CC1 protocol, a statistically significant threshold x epoch x interaction (F_2,145.85_ = 7.65, *p* = 0.001) was found that differed between the groups (*p* = 0.045). Interestingly, in contrast to the EP protocol, NC increased their PT (9.38%) and TT (14.95%, *p* = 0.006) more than in the FM group that showed a PT increase of 6.88% and TT increase of 9.95% (*p* = 0.042, Figure 3).

The CC2 protocol displayed a statistically significant threshold x epoch x interaction (F_2,165.83_ = 8.67, *p* = 0.001) in which the groups differed as a trend (*p* = 0.065) and NC showed a greater increase in PT and TT than the FM group (Figure 3). 

Efficacy of protocols for increasing thresholds: PT and TT in FM patients assessed after the EP protocol were significantly higher than PT and TT after the CC1 protocol (PT: t(31) = 2.675, *p* = 0.021, TT: t(31) = 2.476, *p* = 0.028) and CC2 protocol (PT: t(31) = 3.175, *p* = 0.041, TT: t(31) = 3.174, *p* = 0.006). For NC, the effects of the EP protocol on PT and TT were not significantly different from the effects of either the CC1 or the CC2 protocol.
**2.** **Changes in clinical pain**

The mean VAS rating of clinical pain in FM patients prior to stimulation was 40 (VAS 0–100). The ANCOVA revealed a significant protocol x time interaction (F_2,53_ = 11.92, *p* < 0.001): In the FM patients, the EP protocol resulted in a statistically significant reduction in clinical pain by 35.52% (t (31) = 3.825, *p* = 0.001) after two, ≈8 min stimulations, the CC1 protocol did not show any statistically significant differences, and the CC2 protocol showed a statistically significant increase reported clinical pain by 15.13% (t (31) = −2.105, *p* = 0.042) (Figure 4).
**3.** **Changes in BRS**

Due to a lack of synchrony between BP and HR which is essential for the BRS calculation, values could only be calculated for 51.53% of FM and 53.89% of NC participants. The reduced sample was sufficient for (1) independent sample t-tests of the differences between FM and NC in BRS, and (2) paired-sample t-tests to determine which protocol showed the most statistically significant change in BRS.

FM patients showed statistically significantly lower BRS than NC in baseline, in the first epoch of the EP protocol (*p* = 0.008) and in baseline and in both epochs of the control protocols (CC1 protocol: *p* = 0.023, *p* = 0.022 and CC2 protocol: *p* = 0.009, *p* = 0.007). Importantly, there were no significant differences between FM and NC in the second epoch of the EP protocol.

In FM, the EP protocol resulted in statistically significantly increased BRS between baseline and the second epoch (t(15) = 3.17, *p* = 0.012), as well as between the first and the second epochs (t(15) = 3.163, *p* = 0.019), while the CC1 and CC2 protocols had no effect. For NC, statistically significant changes in BRS were found between the first and second epochs of the CC2 protocol (t(17) = 4.53, *p* = 0.008), but not during the EP or CC1 protocols (Figure 5).
**4.** **Baseline and changes in HRV**

Baseline (pre-stimulation) HRV variables (TP, HF, LF, VLF, SSDN, RMSSD) were statistically significantly lower in FM patients when compared to NCs (all *p* values < 0.003, Table 1 and Figure 6). 

Between-group differences during stimulation showed significantly lower HRV in FM in five of 36 HRV variables compared to NCs: TP during the first epoch of the CC1 protocol (*p* = 0.021), VLF of the first and second epoch of the EP protocol (both *p* < 0.05), LF during the first epoch of CC2 protocol (*p* = 0.037), and RMSSD during the second epoch of the CC2 protocol. During the EP protocol, FM patients were comparable to NCs in their HRV (TP, HF, LF, SDNN and RMSSD), whereby the parasympathetic response (HF) during SP was significantly higher in FM than in NCs (Figure 6).

Within-group differences showed that all HRV variables increased significantly in FM patients across epochs (all *p* values < 0.015) except for VLF during the CC2 protocol. In contrast, only the TP and SDNN during the CC2 protocol (*p* = 0.018 and *p* = 0.016) increased statistically significantly in NCs (Table 1). 

Tests of differences between protocols revealed that in FM, the change in HF from baseline in the EP protocol, indicating parasympathetic activity, was significantly greater than in the two control protocols CC1 and CC2 (both *p* values < 0.01). In contrast, LF and VLF changes from baseline, indicating sympathetic activity, were significantly lower (all *p*’s < 0.03) in the EP protocol in comparison to the control protocols.

## 4. Discussion

The concept of resetting the baroreceptor mechanism that mediates chronic pain is based on both basic (e.g., [6,19,34]) and clinical evidence (e.g., [17,18,24,42]). Dworkin et al. (1991) formulated a theory of learned hypertension in which increased BP and resultant baroreceptor activation reduced stress and pain. This reduction served as a reinforcer that maintains high BP, providing a physiological coping mechanism [6,24]. However, continued exposure to stress dampens fluctuations (variability) in BP, resulting in an adaptation that diminishes the conditioned response and BRS [42,43,44]. 

Animal studies demonstrate analgesia after stimulating the NTS [27,45], the nucleus raphe magnus (NRM, [46,47]), or the central nucleus of amygdala [48,49]. Human studies that experimentally stimulated baroreceptors during the systolic peak of the cardiac cycle demonstrated analgesia in healthy individuals predisposed to hypertension [26,50,51]. The present studies evaluated the possibility of resetting baroreceptor sensitivity in FM. 

*Experimental electrical manipulation of baroreceptors is effective*. In FM, BRS was significantly increased by 41%, clinical pain was reduced by 35% and measures of experimental pain sensitivity were also reduced after two, ≈8 min stimulation sessions that used both pain and pain-free electrical stimuli synchronized to the systolic or diastolic phase of the cardiac cycle.

*Baseline comparison between groups*. Before stimulation, both pain threshold and pain tolerance were significantly lower in FM patients compared to controls, indicating increased pain sensitivity in FM. These results are consistent with those in previous studies using electrical (e.g., [3]), mechanical (e.g., [37]), or thermal (e.g., [52]) pain stimuli [53,54,55]. Neuroimaging studies suggest that decreased pain thresholds and tolerances are related to cortical and/or subcortical augmentation of pain processing [13] and could be related to cerebral midbrain spinal mechanisms of pain inhibition [53].

*The effect of the experimental (EP) and the control (CC1) protocols between groups*. After stimulation with pain and pain-free stimuli during systolic or diastolic cardiac cycle phases (EP protocol), pain threshold and tolerance in FM patients increased by 15.1% and 25.2%, respectively, whereas the 30 pain-free individuals showed significantly smaller increases. In contrast, pain sensitivity was not decreased in the control condition in which FM patients received only painful cardiac-gated stimuli (CC1 protocol). The necessary inclusion of non-painful stimuli for effective analgesia suggests that this combination provokes associative learning by classical conditioning, similar to the need to use both noxious and relaxation stimuli in behavior therapy of anxiety disorders [38]. Such classical conditioning of physiological responses has been shown to influence pain chronicity [56]. Recently published treatment studies showed changes in the prefrontal cortex related to operant and classical conditioning [49,57]. Dependent on cardiac cycle, the combination of pain with pain-free stimuli leads to higher effects in pain inhibition caused by classical conditioning. How this response varies as function of learning is an important subject for future investigation.

*Effect of second control protocol (CC2)*. During the second control condition, painful and non-painful stimuli were applied independently of the cardiac cycle. In contrast to the EP protocol, PT and TT were not increased in FM, validating the hypersensitivity of the central nociceptive system and deficient pain-inhibiting mechanisms involved in the etiology of FM (e.g., [58,59]). Only the EP protocol increased BRS in FM patients. The BRS increase provides important information about the influence of BP and BRS on pain chronicity in FM patients. The BRS modification between groups, or over longer periods of time, indicates a dynamic effect in which arterial baroreceptors modulate the processing of nociception during the cardiac cycle. Several studies have reported altered electro-cortical activity after specific cardiovascular events that alter baroreceptor activity [24,50]. McIntyre [60] found that systolic inhibition of nociceptive responding is moderated by increased central arousal. Painful stimuli given during the systolic phase of the cardiac cycle increase BRS and activate brain regions associated with descending pain inhibition such as the periaqueductal gray [61], the nucleus raphe magnus [36], the NTS [36], and the rostral ventromedial medulla [62,63] that may contribute to the observed pain inhibition.

Furthermore, the unaltered pain thresholds, pain tolerance and BRS after stimulation that is independent of the cardiac cycle validate the effects observed during the synchronized stimulation in the EP protocol and argue against a placebo effect. The CC2 protocol can be considered a sham control condition in this study and represents effects that may occur in real-life conditions.

*Heart rate variability (HRV)*. Consistent with the diminished BRS results, HRV measures of sympathetic and parasympathetic nervous system activity were statistically significantly lower in FM patients than in NCs. The functional consequences of diminished BRS, therefore, seem to include impaired inhibition of sympathetic nervous system arousal responses and the impaired activation of parasympathetic nervous system inhibitory responses evoked by stressful stimuli [19]. The highest parasympathetic activation (HF) and the lowest sympathetic activation (LF) of the nervous system was found after the EP protocol. These results suggest an effect on vagal activity, mediated by the increased BRS that shifts sympathicovagal balance towards parasympathetic tone. The results demonstrate that the changes in heart rate variability, along with an increase of parasympathetic response, suggest an increase of BRS and a reactivation of the dmNTS reflex arcs.

*BRS adaptation*. Since decreased BRS is associated with increased pain, with increased anxiety levels [41] and with both acute [64] and chronic stress [24], the question remains, why does diminished BRS develop in chronic pain patients? Central noradrenergic mechanisms may account for the reduced BRS seen in chronic pain patients [65]. Possible interactions with neurokinin (e.g., substance *p*) and alpha-2 adrenergic pathways within the NTS influence baroreceptor-mediated cardiovascular regulation. Chronic pain-related adaptations in these signaling pathways may contribute to baroreceptor-mediated changes in the BP and pain sensitivity interaction in chronic pain. These specific effects may be driven by learning processes, by so-called “adaptation”, that “refers to the phenomenon whereby baroreceptors activity initially increases with a sustained increase in BP but decline (or adapt) over time as the elevated pressure is maintained” ([43], p. 214).

To summarize, healthy individuals with hypertensive stress reactivity respond to stress with increased BP that reduces levels of stress and pain, reinforcing the increased BP, resulting in hypertension. Pain patients with hypertensive stress response also show a stress-mediated increased BP. However, low BP variability and other factors lead to reduced BRS, NTS vagal activity and pain inhibition. The experience of variable pain and pain-free sensations during both systolic and diastolic phases “resets” BRS. The pronounced clinical effect of reduced pain in a syndrome that defies successful treatment suggests a new and promising method of pain treatment.

The control protocol CC2 was associated with increased sympathetic and decreased vagal activity that likely influence many physiological systems related to *n*. vagus and related diseases and disorders. The effect of diminished BRS and diminished vagal activation due to acute and chronic stress reveals a physiological explanation for the influence of stress on many diseases including chronic pain. In contrast, the EP protocol increased BRS, pain thresholds and pain tolerance resulting in decreased clinical pain. Synchronizing stimulation to the cardiac cycle dramatically altered the effect. The EP protocol is a promising therapeutic intervention for multiple diseases influenced by stress, maladaptive cognitions and pain or disease behaviors.

*Limitations:* Because of the effects of hypotension on BRS [66,67], the sample in this exploratory study consisted of 100% normotensive pain-free controls and 78.6% normotensive and 21.4% hypertensive FM patients. Although EP protocol FM reduced their clinical pain by a third after two, ≈8 min stimulations, future studies are necessary to test this effect in hypertensive FM. In addition, the study was cross-sectional. Future research needs to investigate the stability of the effects observed over time. Finally, the sample consisted of female individuals with FM, only a subset of which were hypertensive. Research is needed to replicate the results obtained with larger samples and with other chronic pain conditions to determine if the results observed were limited to female FM patients or if they are more generalizable to males and other chronic pain conditions.

The EP protocol appears to reset BRS by a mechanism that activates the NTS and, as suggested by increased HRV, vagal response. Further studies are necessary (1) to investigate the interactions between NTS and the affective and cognitive components of the pain network, and (2) to combine the baroreceptor stimulation (EP protocol) with psychological pain therapy that increases the activity of insula and amygdala [57,68,69] associated with clinically significant prolonged pain reduction. In addition, research is needed to understand differences between sample groups of FM patients who do and do not develop hypertension. Finally, it is also important to examine the potential physiological effects achieved with pharmacological interventions to identify the effects of these treatments on BRS-related mechanisms.

## 5. Conclusions

The results implicate BRS/pain sensitivity mechanisms in FM pain chronicity. The positive effects are likely associated with restoring BRS, sympatico-vagal balance, and neuromodulation as altered processing in the pain-related network. Painful and non-painful stimuli during the systolic and diastolic phases of the cardiac cycle decreases clinical pain and pain sensitivity. A preliminary study has demonstrated that the experimental protocol used as a baroreflex training, in combination with operant behavioral therapy, can activate inhibition in various components of the pain network [70]. Restoring BRS, known to be depreciated in FM, and sympatico-vagal balance seem to be linked by classical and operant conditioning. 

The paper provides support for the role of learning in BRS and NTS reflex arcs for pain inhibition as related to increased HRV.

## 6. Patents

US9604054B2 Available online: https://patents.google.com/patent/US9604054B2/en (accessed on 16 October 2022).

## Figures and Tables

**Figure 1 jcm-11-06220-f001:**
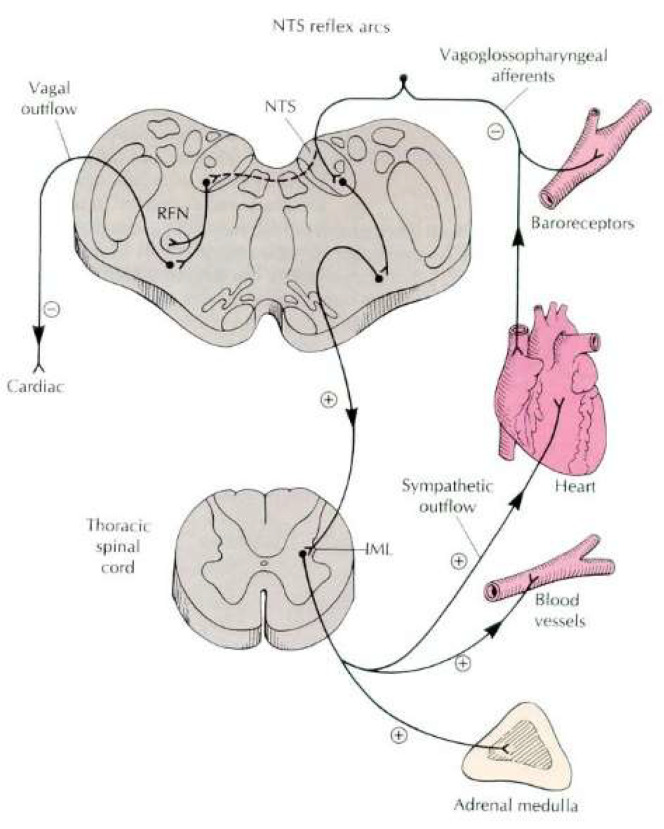
NTS reflex arcs.

**Figure 2 jcm-11-06220-f002:**
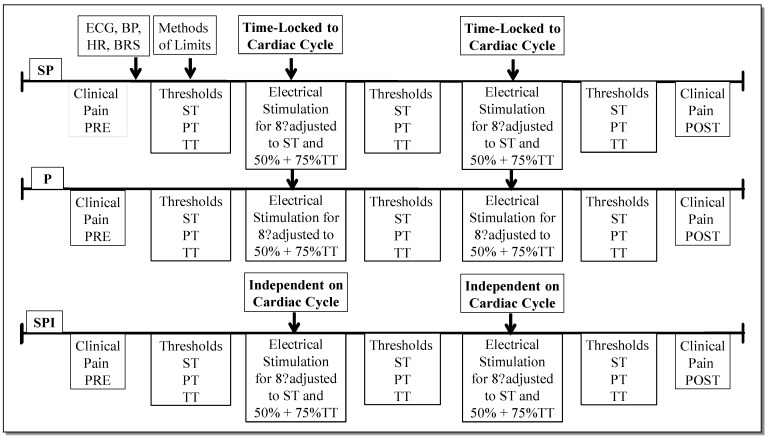
Design of the experiment with the EP, CC1 and CC2 protocols. CC1: control condition 1 protocol. CC2: control condition 2 protocol. EP: experimental protocol. PT: pain threshold. ST: sensory threshold. TT: tolerance threshold.

**Figure 3 jcm-11-06220-f003:**
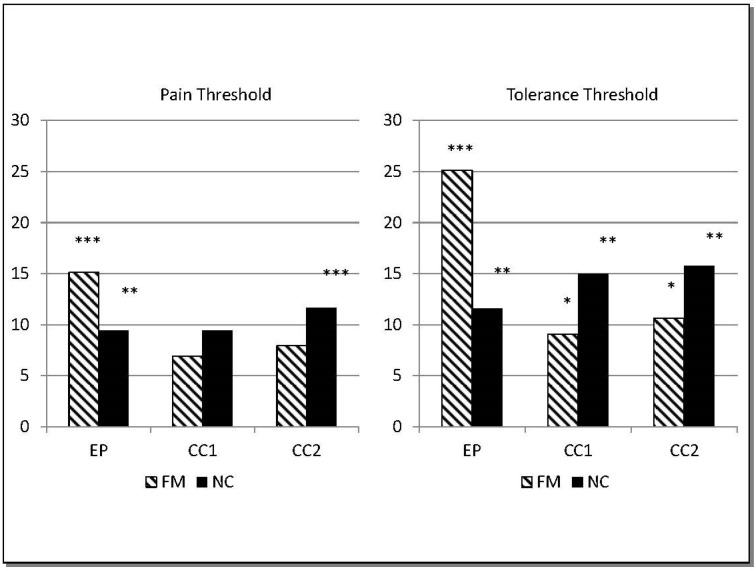
Changes in pain thresholds and tolerance (in %) in response to 2, ≈8 min EP, CC1, and CC2 protocols. * *p* < 0.05, ** *p* < 0.01, *** *p* < 0.001. CC1: control condition 1. CC2: control condition 2. EP: experimental protocol. FM: fibromyalgia patients. NC: normotensive controls.

**Figure 4 jcm-11-06220-f004:**
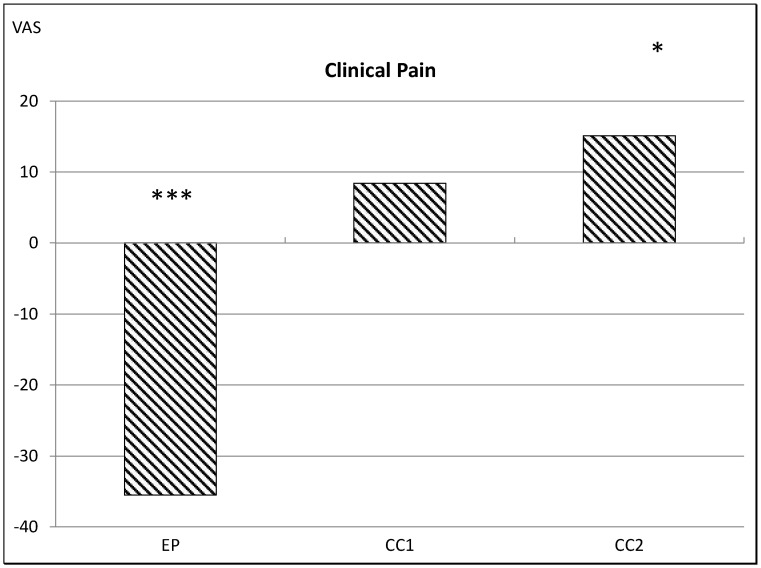
Changes in clinical pain intensity in response to 2, ≈8 min EP, CC1 and CC2 protocols assessed by visual analogue scale (0–100). * *p* < 0.05, *** *p* < 0.001 comparing pre- vs. post-responses. CC1: control condition 1. CC2: control condition 2. EP: experimental protocol.

**Figure 5 jcm-11-06220-f005:**
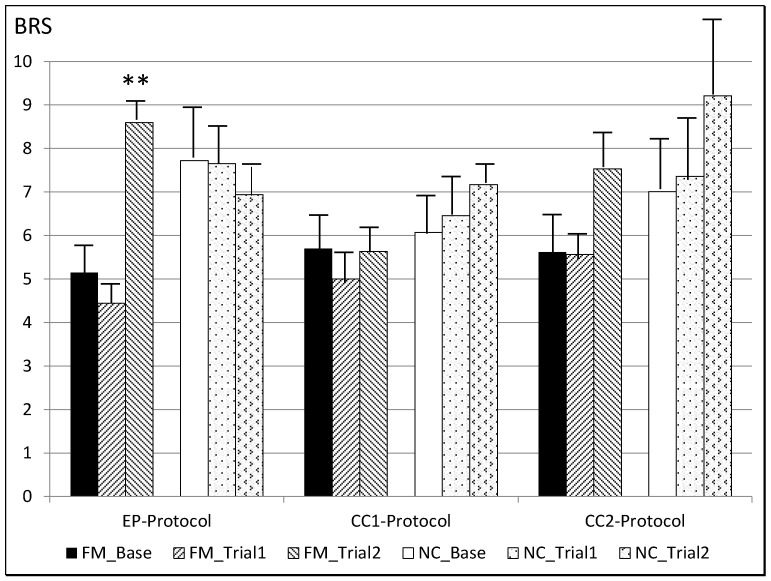
Mean BRS values in EP, CC1, and CC2 protocols, ** *p* < 0.01 comparing baseline to epoch 1st and to epoch 2nd in FM and NC. Base: baseline. BRS: baroreflex sensitivity. CC1: control condition 1. CC2: control condition 2. EP: experimental protocol. FM: fibromyalgia patients. NC: normotensive controls.

**Figure 6 jcm-11-06220-f006:**
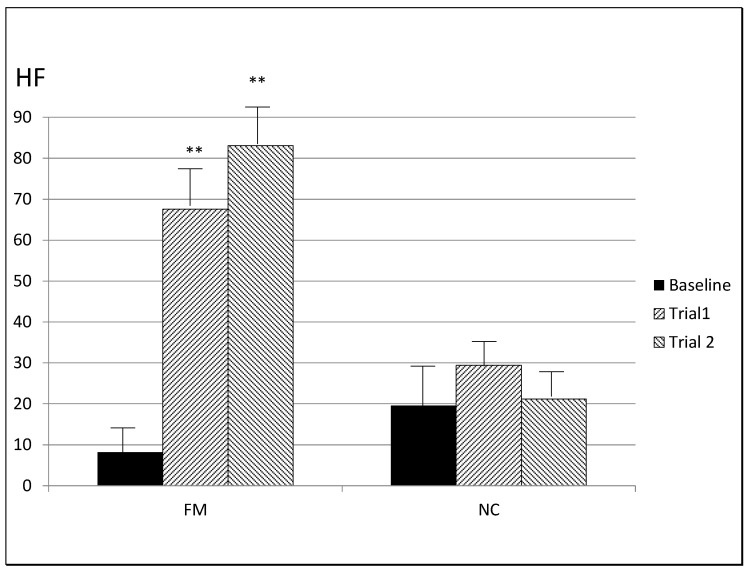
Vagal activity in baseline, epoch 1 and epoch 2 of EP protocol measured by HF in FM and NC. ** *p* < 0.01 comparing baseline to epoch 1 and to epoch 2. FM: fibromyalgia patients. HF: high frequency. NC: normotensive controls.

**Table 1 jcm-11-06220-t001:** Suppl. natural log mean and SD’s values of different HRV variables in FM patients and NC, differences between and within the groups during 5 min baseline, ≈8 min epoch 1 and 2.

	Groups		Sign Groups		Sign FM		Sign NC	
Variable	FM Mean (SD)	NC Mean (SD)	U	*p*	Chi2	*p*	Chi2	*p*
**TP**								
Baseline	5.31 (1.10)	6.62 (0.95)	171	<0.001				
EP_1	6.48 (0.89)	6.86 (0.76)	240	ns				
EP_2	6.58 (0.86)	6.72 (0.81)	190	ns	24.78	<0.001	2.79	ns
CC1_1	6.43 (0.77)	6.82 (0.72)	153	0.021				
CC1_2	6.62 (0.96)	6.63 (0.67)	203	ns	9.38	0.009	0.22	ns
CC2_1	6.40 (0.74)	6.79 (0.72)	160	ns				
CC2_2	6.62 (0.79)	6.82 (0.83)	183	ns	13.85	0.001	8.07	0.018
**HF**								
Baseline	2.08 (0.79)	3.06 (0.93)	188	0.001				
EP_1	3.08 (1.38)	3.11 (0.78)	298	ns				
EP_2	2.98 (1.62)	2.94 (0.75)	247	ns	9.44	0.014	2.64	ns
CC1_1	2.63 (0.56)	3.33 (1.06)	177	ns				
CC1_2	2.65 (0.62)	3.08 (0.89)	152	ns	11.69	0.003	0.67	ns
CC2_1	2.62 (0.59)	3.15 (0.89)	172	ns				
CC2_2	2.87 (0.83)	3.21 (1.03)	161	ns	13.86	0.001	0.67	ns
**LF**								
Baseline	3.91 (1.13)	5.03 (1.18)	218	0.003				
EP_1	4.96 (0.84)	4.98 (0.87)	296	ns				
EP_2	5.18 (1.05)	5.01 (0.85)	239	ns	14.11	0.001	0.07	ns
CC1_1	4.64 (0.85)	5.02 (0.83)	161	ns				
CC1_2	4.92 (1.02)	4.97 (0.84)	184	ns	8.77	0.012	0.52	ns
CC2_1	4.66 (1.01)	5.10 (0.99)	153	0.037				
CC2_2	4.82 (0.92)	5.09 (1.02)	186	ns	16	<0.001	2	ns
**VLF**								
Baseline	4.93 (0.83)	6.07 (0.86)	184	<0.001				
EP_1	5.64 (0.92)	6.25 (0.77)	189	0.02				
EP_2	5.81 (0.91)	6.16 (0.83)	158	0.034	8.44	0.015	2.214	ns
CC1_1	6.03 (0.80)	6.15 (0.74)	209	ns				
CC1_2	5.97 (0.88)	6.05 (0.71)	187	ns	9.39	0.009	0.89	ns
CC2_1	5.90 (0.79)	6.21 (0.76)	162	ns				
CC2_2	6.02 (0.86)	6.24 (0.84)	181	ns	6.14	ns	4.52	ns
**SDNN**								
Baseline	2.73 (0.42)	3.31 (0.47)	164	<0.001				
EP_1	3.26 (0.44)	3.42 (0.37)	274	ns				
EP_2	3.29 (0.41)	3.35 (0.39)	227	ns	21.38	<0.001	2	ns
CC1_1	3.22 (0.38)	3.41 (0.35)	175	ns				
CC1_2	3.30 (0.47)	3.31 (0.34)	225	ns	9.38	0.009	0.96	ns
CC2_1	3.22 (0.37)	3.39 (0.36)	157	ns				
CC2_2	3.30 (0.39)	3.41 (0.42)	185	ns	13.29	0.001	8.29	0.016
**rmSSD**								
Baseline	1.68 (0.35)	2.19 (0.44)	208	0.002				
EP_1	2.31 (0.77)	2.20 (0.37)	284	ns				
EP_2	2.39 (0.81)	2.17 (0.33)	236	ns	16.63	<0.001	1.14	ns
CC1_1	2.66 (1.71)	2.34 (0.52)	184	ns				
CC1_2	2.66 (1.68)	2.24 (0.45)	201	ns	22.8	<0.001	0.29	ns
CC2_1	2.65 (1.75)	2.24 (0.40)	207	ns				
CC2_2	1.99 (0.34)	2.29 (0.54)	152	0.035	16.13	<0.001	1.185	ns

CC1_1: control condition 1–protocol epoch 1. CC1_2: control condition 1–protocol epoch 2. CC2_1: control condition 2–protocol epoch 1. CC2_2: control condition 2–protocol epoch 2. EP_1: experimental protocol epoch 1. EP_2: experimental protocol epoch 2, HF: High Frequency. LF: Low Frequency. ns: no significant. RMSSD: root mean square of the differences between successive N–N intervals. Sign FM: Significant Differences within Fibromyalgia. Sign Groups: Significant Differences between the groups. Sign NC: Significant differences within normotensive controls. SSDN: standard deviation of normal-to-normal [N–N] intervals. TP: Total Power, VLF: Very Low Frequency.

## Data Availability

Data available on request due to restrictions e.g., privacy or ethical.

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
