# Peer review of "Cardiac-Gated Neuromodulation Increased Baroreflex Sensitivity and Reduced Pain Sensitivity in Female Fibromyalgia Patients"

_jcm, 2022, doi:10.3390/jcm11206220_

Round 1
Reviewer 1 Report
This topic is very interesting, but paper needs some revisions. Look at these points:
- Lines 53-54: "In contrast, in FM and other chronic pain patients..." Which others chronic pain ? Improve.
- Lines 38-40: "BP elevations.. vessel dilation, autonomic balance, sleep, and pain perception (Figure 1)[6-8]." Look at more recent papers. Consider these 2 important refs: -- Scarola et al. Temporomandibular Disorders and Fibromyalgia: A Narrative Review. Open Access Maced J Med Sci [Internet]. 2021; 9 (F):106-12. doi: 10.3889/oamjms.2021.5918 --- Gerdle et al. Fibromyalgia: Associations Between Fat Infiltration, Physical Capacity, and Clinical Variables. J Pain Res. 2022 Aug 27;15:2517-2535.
- Lines 73-74: "We hypothesize that electrical stimuli delivered during specific phases of the cardiac cycle will... " Please state better what is the aim of your paper.
- Lines 273-278: "3.1. Subsection.." Improve this part.
- Figure 4. Improve figure legends.
- Lines 399-404: "Baseline comparison between groups. Before... using electrical (e.g., 3), mechanical (e.g., [37]), and thermal (e.g., [53]) pain stimuli [54-56]" Are these results?
- Lines 446-448: "The 446 highest activation of the parasympathetic nervous... of the sympathetic nervous system (LF) were found after the SP protocol." What do authors want to say with this sentence?
- Lines 417-419: "Differences in brain responses between electrical stimulation with and without classical conditioning of pain stimuli after systolic peak are important targets for future studies" Improve this topic.
- Conclusion should be improved. What this paper add new to the literature?
Author Response
Responses to the Reviewer 1
This topic is very interesting, but paper needs some revisions. Look at these points:
- Lines 53-54: "In contrast, in FM and other chronic pain patients..." Which others chronic pain ? Improve.
>>>>>>>>>>>> We very appreciate the valuable suggestions of the reviewer. We included “orofacial pain, migraine, rheumatoid arthritis” as well as the references 7 and 18 (lines 53 and 55).
In contrast, in FM, orofacial pain, migraine, and rheumatoid arthritis, baroreflex sensitivity (BRS) is reduced and the responses to stress and pain are blunted [7,18,22]
- Lines 38-40: "BP elevations.. vessel dilation, autonomic balance, sleep, and pain perception (Figure 1)[6-8]." Look at more recent papers. Consider these 2 important refs: -- Scarola et al. Temporomandibular Disorders and Fibromyalgia: A Narrative Review. Open Access Maced J Med Sci [Internet]. 2021; 9 (F):106-12. doi: 10.3889/oamjms.2021.5918 --- Gerdle et al. Fibromyalgia: Associations Between Fat Infiltration, Physical Capacity, and Clinical Variables. J Pain Res. 2022 Aug 27;15:2517-2535.
>>>>>>>>>>>>>>We included the 2 papers that the reviewer has recommended (line 40) and have changed the numbers of the references in the manuscript using endnote.
- Lines 73-74: "We hypothesize that electrical stimuli delivered during specific phases of the cardiac cycle will... " Please state better what is the aim of your paper
>>>>>>>>>>>>>We changed the original sentence: We hypothesize that electrical stimuli delivered during specific phases of the cardiac cycle will reset BRS, regulate HRV and reduce pain sensitivity and clinical pain in a subgroup of FM patients characterized by normotensive resting BP.
To (lines 75-78)
We hypothesize that electrical stimuli delivered during systolic and diastolic phases of the cardiac cycle in a randomized order would reset BRS, regulate HRV and reduce pain sensitivity and clinical pain in a subgroup of FM patients characterized by normotensive resting BP.
- Lines 273-278: "3.1. Subsection.." Improve this part.
>>>>>>>>>>>> We have improved the part:
- Results.
This section may be divided by subheadings. It should provide a concise and precise description of the experimental results, their interpretation, as well as the experimental conclusions that can be drawn.
3.1. Subsection
3.1.1. Subsubsection
- Changes in Threshold;
- Changes in Clinical Pain;
- Changes in BRS;
- Changes in Heart Rate Variability.
To (lines 281-282)
In the following paragraph, we present the changes in (1) Thresholds, (2) Clinical Pain, (3) BRS and (4) Heart Rate Variability.
- Figure 4. Improve figure legends.
>>>>>>>>>>> We improved the legends of Figure 4
Figure 4. Changes in clinical pain intensity in response to 2, 8-minute EP, CC1 and CC2 protocols assessed by visual analogue scale (0 – 100).
* P < 0.05, *** P < 0.001 comparing pre vs post responses.
CC1, Control Condition 1- Protocol, CC2, Control Condition 2 - Protocol EP, Experimental Protocol.
To (lines 335-6)
Figure 4. Changes in clinical pain intensity in response to EP, CC1 and CC2 protocols assessed by numeric rating scale (0-100) before and after each protocol.
* P < 0.05, *** P < 0.001 comparing pre vs post responses.
CC1, Control Condition 1- Protocol, CC2, Control Condition 2 - Protocol EP, Experimental Protocol.
- Lines 399-404: "Baseline comparison between groups. Before... using electrical (e.g., 3), mechanical (e.g., [37]), and thermal (e.g., [53]) pain stimuli [54-56]" Are these results?
>>>>>>>>>>>>>>>>>>>The paragraph belongs to the discussion and is related to the results written in the Lines 279-283:
Baseline sensory, pain and tolerance thresholds. In contrast to comparable sensory thresholds in FM patients and NC group, pain threshold (F(1;69) = 20.36, p < 0.001) and tolerance (F(1;69) = 28.75, p < 0.001) values differed significantly. FM patients showed a 12.89% lower pain threshold value and a 25.85% lower tolerance value than NC’s.
For a better understanding, we changed
Baseline comparison between groups. Before stimulation, both pain threshold and pain tolerance were significantly lower in FM patients compared to controls, indicating increased pain sensitivity in FM. These results are consistent with those in previous studies using electrical (e.g. 3), …
To (line 418)
Baseline comparison between groups. Before stimulation, both pain threshold and pain tolerance were significantly lower in FM patients compared to controls, indicating increased pain sensitivity in FM. Our results are consistent with those in previous studies using electrical (e.g.3), …
- Lines 446-448: "The 446 highest activation of the parasympathetic nervous... of the sympathetic nervous system (LF) were found after the SP protocol." What do authors want to say with this sentence?
>>>>>>>>>>>>>To improve the paragraph: “The highest activation of the parasympathetic nervous system (HF) and the lowest activation of the sympathetic nervous system (LF) were found after the SP protocol. These results suggest an effect on vagal activity mediated by the increased BRS that shifts sympathicovagal balance towards parasympathetic tone. “
we added on lines 467-469 :
The results demonstrate that the changes in heart rate variability along with an increase of parasympathetic response suggest an increase of BRS and a reactivation of the dmNTS reflex arcs.
- Lines 417-419: "Differences in brain responses between electrical stimulation with and without classical conditioning of pain stimuli after systolic peak are important targets for future studies" Improve this topic.
>>>>>>>>>>>>>>>We improved the sentence (lines 436-39)
Dependent on cardiac cycle, the combination of pain with pain-free stimuli leads to higher effects in pain inhibition caused by classical conditioning. How this response varies as function of learning is an important subject for future investigation.
- Conclusion should be improved. What this paper add new to the literature?
We added on line 529-530:
The paper provides support for the role of learning in BRS and NTS reflex arcs for pain inhibition as related to increased HRV.
Thank you very much for your support!

Reviewer 2 Report
Kati Thieme and colleagues presented an interesting study on female fibromyalgia patients. The results seem sound. However, the manuscript is hard to read. It should be written more concise, and the language has to be improved. It is not necessary to refer to 71 papers, 7 of those are from the first author.
My major question is how the authors controlled the placebo effect in their experiment. Patients with fibromyalgia have neuropsychiatric comorbidities, and part of the pain relief may be due to placebo.
Author Response
Responses to Reviewer 2
Kati Thieme and colleagues presented an interesting study on female fibromyalgia patients. The results seem sound. However, the manuscript is hard to read. It should be written more concise, and the language has to be improved. It is not necessary to refer to 71 papers, 7 of those are from the first author.
>>>>>>>>>There are 4 references from the first author. We have deleted 2 references:
Diers M, Yilmaz P, Rance M, Thieme K, Gracely RH, Rolko C, Schley MT, Kiessling U, Wang H, F. H., Treatment-related changes in brain activation in patients with fibromyalgia syndrome., Exp Brain Res, 218 (2012) 619-628.
Pertovaara, A. A neuronal correlate of secondary hyperalgesia in the rat spinal dorsal horn is submodality selective and facilitated by supraspinal influence., Exp Neurol 149 (1998) 193-202.
My major question is how the authors controlled the placebo effect in their experiment. Patients with fibromyalgia have neuropsychiatric comorbidities, and part of the pain relief may be due to placebo.
>>>>>>>>>The patients were investigated with SCID I and II to control neuropsychiatric conditions. Patients with personality disorders and severe depression were excluded. We controlled a possible placebo effect by the Control Condition 2 (CC2) in which pain-free and pain stimuli were applied independent on cardiac cycle. The intensity of the stimuli in CC2 were comparable to the EC. The conditions applied the stimuli in different time spaces of the cardiac cycle. While the stimuli in CC2 were applied in the middle (50%) of the cardiac cycle, the stimuli in EC were randomized applied in 20% (after systolic peak) and 80% (after diastolic peak).
I have attached also our responses to the reviewer 1. Thus, you can see what we have changed to follow your suggestion to make the manuscript more concise.
Thank you very much for your support and your impact.

Round 2
Reviewer 1 Report
Authors solved all my criticisms